# Can Pharmaceutical Excipients Threaten the Aquatic Environment? A Risk Assessment Based on the Microtox^®^ Biotest

**DOI:** 10.3390/molecules28186590

**Published:** 2023-09-13

**Authors:** Marika Turek, Ewa Różycka-Sokołowska, Marek Koprowski, Bernard Marciniak, Piotr Bałczewski

**Affiliations:** 1Institute of Chemistry, Faculty of Science and Technology, Jan Długosz University in Częstochowa, Armii Krajowej 13/15, 42-200 Częstochowa, Poland; e.sokolowska@ujd.edu.pl (E.R.-S.); b.marciniak@ujd.edu.pl (B.M.); 2Division of Organic Chemistry, Center of Molecular and Macromolecular Studies, Polish Academy of Sciences, Sienkiewicza 112, 90-363 Łódź, Poland; mkopr@cbmm.lodz.pl

**Keywords:** aquatic toxicity, cardiovascular drugs, ecotoxicity, excipients, Microtox^®^, mixture toxicity, toxicity prediction, CA model

## Abstract

The ecotoxicological impact of pharmaceuticals has received considerable attention, primarily focusing on active pharmaceutical ingredients (APIs) while largely neglecting the potential hazards posed by pharmaceutical excipients. Therefore, we analyzed the ecotoxicity of 16 commonly used pharmaceutical excipients, as well as 26 API–excipient and excipient–excipient mixtures utilizing the Microtox^®^ test. In this way, we assessed the potential risks that pharmaceutical excipients, generally considered safe, might pose to the aquatic environment. We investigated both their individual ecotoxicity and their interactions with tablet ingredients using concentration addition (CA) and independent action (IA) models to shed light on the often-overlooked ecotoxicological consequences of these substances. The CA model gave a more accurate prediction of toxicity and should be recommended for modeling the toxicity of combinations of drugs with different effects. A challenge when studying the ecotoxicological impact of some pharmaceutical excipients is their poor water solubility, which hinders the use of standard aquatic ecotoxicity testing techniques. Therefore, we used a modification of the Microtox^®^ Basic Solid Phase protocol developed for poorly soluble substances. The results obtained suggest the high toxicity of some excipients, i.e., SLS and meglumine, and confirm the occurrence of interactions between APIs and excipients. Through this research, we hope to foster a better understanding of the ecological impact of pharmaceutical excipients, prompting the development of risk assessment strategies within the pharmaceutical industry.

## 1. Introduction

Most research on pharmaceuticals in the environment concerns active pharmaceuticals ingredients (APIs); however, it should be remembered that, in order to produce a pharmaceutical product with the desired physicochemical properties, a number of auxiliary substances (pharmaceutical excipients) have to be used, i.e., surfactants, lubricants, emulsifiers, and preservatives. Many of these substances are also used in foods and cosmetics. Considering the number of drugs taken and the fact that each pharmaceutical preparation contains, on average, from a few to a dozen or even more excipients, an assessment of the risks associated with the presence of these substances in the environment seems reasonable [1]. Currently, US and European Union regulations require auxiliary substances to be toxicologically tested and meet quality standards, but no environmental risk assessment is required, as is the case for APIs [1].

For example, in Sweden, for the 7600 pharmaceutical products available for sale, there are as many as 1300 different excipients [2]. According to the report from 2019, excipients constitute on average 71% of the weight of a tablet; therefore, they predominate the weight of finished pharmaceutical formulations [3]. The same report, which was based on the Pillbox database, showed that the average tablet/capsule contains 8.8 excipients, with some even containing up to 34 excipients. Moreover, different amounts of excipients are used for the same API depending on the manufacturer. The most common excipient in tablets, used in 72% of cases, is magnesium stearate, one of the most popular lubricants supporting technological processes.

Some of the excipients, such as sugars, starch, cellulose, proteins, or glycerol, are generally considered safe for the environment. Other substances, such as preservatives, cause more concern due to their anti-microbial properties [4]. Similarly, surfactants may have some biological effects on organisms in the environment [5]. There are only a few studies available in the scientific literature regarding the ecotoxicity of pharmaceutical excipients. These are usually studies on the ecotoxicity of various commercial pharmaceutical forms of APIs that contain different compositions of excipients. On the basis of this type of research, it can be concluded that excipients may increase the toxicity of APIs, possibly through interactions with the API itself or through interactions of different excipients with each other [6,7]. It is worth mentioning that similar studies were carried out with regard to commercial herbicide formulations, and it was found that auxiliary substances may be more ecotoxic than the active substance itself [8,9]. Currently, little is known about the ecotoxicity of pharmaceutical excipients. Although these substances are present in relatively small amounts in a single pharmaceutical formulation and are potentially safe substances, it is suggested that the environmental risk assessment should consider (1) their total amount in the environment, proportional to the volume of production, and (2) the entire pharmaceutical formulation, including APIs and excipients, rather than individual ingredients [6]. Unfortunately, the number of studies in this area is negligible. 

There is one important study regarding the ecotoxicity of entire drug formulations and the influence of excipients on toxicity of API and final formulation. This research utilized the Microtox^®^ test to evaluate ecotoxicity of various drug formulations (generic and original) of 10 popular APIs [7]. It is worth mentioning that the Microtox^®^ test is based on the bioluminescence inhibition of the *Allivibrio fischeri* bacterium, and it is a widely used ecotoxicity bioassay for assessing the toxicity of various compounds or solid matrices (such as soils) to aquatic organisms. The results obtained by the authors showed that the ecotoxicological risk is not necessarily related only to the presence of APIs in the environment, but may also be related to the excipients contained in the tablets. On the basis of the comparative analysis of toxicity of formulations containing various excipients, it could be concluded that the excipients that may increase the toxicity of the drug formulation are microcrystalline cellulose, sodium starch glycolate, silicon dioxide, and SICOVIT-10 BASF yellow dye [7]. Other researchers also mentioned that the potentially toxic effect associated with excipients is often overlooked [2,6].

Therefore, in our previous work, we paid attention to the occurrence of API–API, API–excipient, and excipient–excipient antagonistic/synergistic interactions based on the example of poorly soluble antihypertensive angiotensin II receptor blockers (ARBs) [10]. The study of these interactions was possible thanks to the developed modification of the standard Microtox^®^ Basic Solid Phase Test protocol, which allowed us to reliably assess the ecotoxicity of poorly soluble APIs in tests of both individual compounds and mixtures. This modified protocol is legally protected by WIPO Patent Application WO/2021/215944 (22 April 2021).

As a continuation of our interest, the aim of this work was to evaluate the ecotoxicity of individual excipients used in ARB pharmaceutical formulations, as well as the ecotoxicity of various ARB–excipient and excipient–excipient mixtures including multi-component mixtures (Figure 1). The experimental ecotoxicity of the mixtures was compared with the ecotoxicity predicted using common models: CA (concentration addition) and IA (independent action). It is believed that the CA model is suitable for predicting the toxicity of mixtures of similarly acting chemicals. The IA model, unlike the CA model, is based on the assumption that the components of the mixture act differently on the test organism, i.e., they produce a biological effect at the molecular level by interacting with different target sites of the organism. Both models are useful for assessing the ecotoxicological risk associated with the presence of mixtures of various chemical compounds in the environment. Acute toxicity is tested with bioluminescent marine bacterium *Allivibrio fischeri* (termed *Vibrio fischeri* until 2007) by utilizing a modification of the standard Microtox^®^ Basic Solid Phase Test protocol, which provides direct contact of the test bacterium with a solid sample and reflects the real situation in the environment where non-target species are exposed to the xenobiotics, which can be dissolved, suspended, and/or adsorbed in the solid matrix [10]. So far, this test is a worldwide standard for acute toxicity testing of water and is also one of the techniques that can be used in water quality biomonitoring in accordance with European standards.

## 2. Results and Discussion

### 2.1. Ecotoxicity of Individual Pharmaceutical Excipients

Results of the ecotoxicity studies of individual excipients using the Microtox^®^ Basic Solid Phase Test are listed in Table 1. Additionally, in order to check, how the particle size of silica, used as an excipient in pharmaceutical formulations, affects the toxicity toward *A. fischeri*, the ecotoxicity of three forms of silica was tested: a nanopowder, a colloidal form (LUDOX SM-30), and colloidal anhydrous silica (Aerosil R200).

Most of the tested excipients commonly used in pharmaceutical formulations, according to the Globally Harmonized System of Classification and Labeling of Chemicals (GHS) [11], are non-toxic (EC_50_ values above 100 mg/L). However, the obtained values among non-toxic compounds are differentiated from several hundred mg/L to even tens of thousands of mg/L; therefore, it is also necessary to carefully consider some substances that are at the threshold of this limit. Two of the analyzed excipients could be classified into III toxicity category (harmful substances)—meglumine and magnesium carbonate—with EC_50_ values in the range of 10–100 mg/L. A worrying result is the toxicity of sodium lauryl sulfate (SLS); this compound can be characterized as very toxic, with an EC_50_ value below 1 mg/L.

#### 2.1.1. Sodium Lauryl Sulfate (SLS)—Toxicity Category I

SLS is used in pharmacy as an anionic surfactant, emulsifier, solubilizer, lubricant, and a modified-release formulation agent [12]. This compound is approved as an excipient in preparations for external use, toothpastes, shampoos, and tablets, in amounts of 0.2–2%, even up to 25% in shampoos [13]. Like other investigated excipients, it is classified as a safe compound according to the FDA GRAS (Generally Recognized As Safe) list. As it is a surfactant, such high toxicity to *A. fischeri* can most likely be explained by the bacteriostatic properties of SLS and its ability to create pores in bacterial membranes through which the cytoplasm content can flow out, causing cell death. Although the literature sources suggest the main activity of SLS is against Gram-positive bacteria, as a result of the conducted studies, it can be concluded that it also has antibacterial properties against Gram-negative bacterium *A. fischeri* [12,14]. Taking into account the large scale of SLS production and the number of its applications in the pharmaceutical and cosmetic industries, this result is alarming from the point of view of environmental safety. The toxicity of SLS has been a subject of many scientific publications and has also been a subject of discussion in the public sphere [15]. With regard to the published ecotoxicological data for various aquatic organisms, the LC_50_ (lethal concentration) values for SLS are in the range of 1–13.9 mg/L [16,17,18,19,20]; thus, this compound can be classified into toxicity class II. However, it is worth remembering that the toxicity values of individual compounds for aquatic organisms do not directly correspond to the toxicity of final pharmaceutical or cosmetic products. For example, the potentially toxic SLS, when used at some dilution in a formulation, may be non-toxic to aquatic organisms. It is also important to remember that synergistic/antagonistic interactions of individual substances may be present in the final pharmaceutical product.

#### 2.1.2. Meglumine and Magnesium Carbonate—Toxicity Category III

Two of the tested excipients—meglumine and magnesium carbonate (heavy)—can be classified into aquatic toxicity category III (compounds harmful to the aquatic environment). Meglumine is a 1-methylamine derivative of sorbitol, used in pharmacy as a pH-adjusting agent and solubilizer. This substance is on the GRAS list of substances and is approved for use as an excipient in capsules, tablets, and injections [21]. However, there are no studies on this compound regarding its ecotoxicity. There are studies on animal models suggesting that meglumine, taken in high doses, may have therapeutic properties in the treatment of diseases, such as obesity, diabetes, and nonalcoholic steatohepatitis [21,22]. As mentioned earlier, meglumine is a derivative of sorbitol; hence, one would expect a similar ecotoxicity of the two compounds. In the course of the conducted studies, it was shown that sorbitol was completely non-toxic at the tested concentration, while the EC_50_ value for meglumine was only 18.05 mg/L. These compounds differ in their chemical structure only in a methylamine group, which gives meglumine basic properties that are unfavorable to the *A. fischeri* bacterium. Such properties are not exhibited by sorbitol, which, under measurement conditions, causes the phenomenon of hormesis, i.e., beneficial effects at low concentrations [23]. The second test compound in this toxicity category III is heavy magnesium carbonate, consisting of MgCO_3_ with 40–50% Mg(OH)_2_ [24]. It is approved for use in pharmaceutical formulations (tablets, capsules) in amounts up to 45% by weight [25]. It acts as an absorbent and a stabilizer [13]. The high toxicity of this compound is probably related, as in the case of meglumine, to the alkaline pH of magnesium carbonate (Appendix A), which is associated with unfavorable conditions for the bacterium.

#### 2.1.3. Microcrystalline Cellulose and Anhydrous Colloidal Silica

In our previous studies [10], microcrystalline cellulose and anhydrous colloidal silica were selected as compounds potentially causing an increase in the toxicity of pharmaceutical formulations. As a result of further studies, it was found that microcrystalline cellulose with an EC_50, 30 min_ value at the level of 1958 mg/L and anhydrous colloidal silica with EC_50, 30 min_ = 2681 mg/L were non-toxic to the environment, although their EC_50_ values were not as high as for other non-toxic excipients investigated in this study. Microcrystalline cellulose is a partially depolymerized form of cellulose obtained from an α-cellulose precursor. The tested microcrystalline cellulose of the PH-102 type has an average particle size of 100 µm and is used in pharmacy as a binding, disintegrating, absorbing, filling, lubricating, and non-sticking agent [26]. As cellulose is a natural component of plants and occurs in the environment, it is assumed that this substance is safe for the environment [27]; however, there are no ecotoxicological studies regarding this compound in the scientific literature. It is suspected that microcrystalline cellulose, due to its small size, may exhibit increased absorption properties and enter cells through its pores, where it bioaccumulates and disrupts cell functioning. In 1997, the European Commission issued a directive banning microcrystalline cellulose particles smaller than 5000 nm (5 µm) [28].

#### 2.1.4. Colloidal Anhydrous Silica—The Effect of Particle Size and Commercial Form

Colloidal anhydrous silica is used in pharmacy as an anti-caking agent, emulsion stabilizer, lubricant, suspending agent, stabilizer, and viscosity-enhancing agent [13]. In this paper, ecotoxicological studies were conducted on Aerosil R200, the form of silica used in the pharmaceutical industry, containing aggregated colloidal silica with a primary particle size of approximately 12 nm [29]. Additionally, to determine the effect of particle size and form of colloidal silica on toxicity to bacterium *A. fischeri*, the ecotoxicity of two other forms of silica was investigated: the liquid form of LUDOX SM-30 (particle size 7 nm) and the nanopowder (particle size 10–15 nm). Taking into account the fact that LUDOX SM-30 contains 30 wt.% SiO_2_, its EC_50, 30 min_ value of 3794 mg/L based on pure SiO_2_ can be considered equal to 1138.2 mg/L. Table 2 summarizes the EC_50_ values toward *A. fischeri* of the various forms of silica along with their particle size.

So far, silica has been the subject of many studies, mainly due to the fact that SiO_2_ nanoparticles are among the most widely produced nanomaterials [32,33]. They find their application in ceramic, glass, and cosmetic products, as well as in medicine and pharmacy [34,35]. Colloidal silica, e.g., Aerosil R200, is used in pharmacy as a lubricant for the production of powders, capsules, and tablets [36,37]. Due to the small size of primary particles and agglomerates, they are strongly adsorbed on the surfaces of larger particles [38]. As silicas are used in the form of nanoparticles, studies related to the safety of these types of compounds, as well as their ecotoxicity, have been widely discussed in the scientific literature. Safekordi et al. investigated the ecotoxicity of two types of silica, i.e., powder and mesoporous silica, toward *A. fischeri*, obtaining similar EC_50_ values for both forms (Table 2) [30]. It is worth noting that the authors did not use the Microtox^®^ apparatus, and the measurements were based on the luminometer developed by the authors. In turn, Casado et al. tested two forms of silica with different particle sizes using the Microtox^®^ protocol for aqueous extracts of the toxicant tested [31]. They showed that the tested forms of SiO_2_ are non-toxic at the maximum tested concentration of 1000 mg/L. The results, shown in Table 2, are for comparison purposes. However, it should be remembered that they were not all obtained using the same test protocol; hence, the quantification of the reported results may be somewhat misleading. In relation to the results obtained in this study, a much higher toxicity of native, powdered silica than the forms used commercially in pharmacy (Aerosil R200) can be noted. This is probably due to the greater availability of SiO_2_ nanopowder for bacteria because it does not form agglomerates or colloidal systems. Although Aerosil R200 has a similar particle size of primary SiO_2_ (12 nm) to the tested powder (10–15 nm), it is 10 times less toxic than the powder. This is due to the agglomeration of silica in the hydrophilic formulation, which is associated with its lower availability for bacteria. In contrast, the LUDOX SM-30 formulation, despite having silica with the smallest particle size (7 nm), is five times less toxic than SiO_2_ nanopowder. This is probably related to the colloidal properties of LUDOX and, therefore, the lower availability of SiO_2_ for *A. fischeri*. LUDOX was also tested by the van Hoecke et al. [39] for its ecotoxicity toward green algae, where this form of silica has been shown to be toxic. The authors of the study found no uptake of SiO_2_ particles into cells, but instead their adsorption onto the cell walls of the test organisms. The effect of SiO_2_ with particle sizes of 20 and 50 nm on aquatic *Daphnia magna* was also investigated. The authors of the study demonstrated no acute toxicity of the tested silica at concentrations up to 100 mg/L. However, significant physiological and behavioral changes were observed in adults and young *D. magna* animals, such as changes in swimming efficiency, individual size, and reproductive impairment; these factors supported the hypothesis that SiO_2_ had a negative effect on *D. magna* [40]. Other authors indicated that both amorphous and crystalline forms of SiO_2_ could lead to lymph-node fibrosis in rats [41]. However, it is worth noting that the predicted concentration of SiO_2_ nanoparticles in the aquatic environment is 562 ng/L (northern Europe) and 2600 ng/L (southeastern Europe); these concentrations are significantly higher than the predicted environmental concentrations of other nanoparticles, i.e., Al_2_O_3_, CeO_2_, and iron oxides, but still much lower than the EC_50_ values obtained for the various forms of silica [42].

#### 2.1.5. Povidones

It is worth paying attention to the toxicity of three compounds in the povidone group—povidone K25, povidone K30, and crospovidone (type A) (Table 1). These are synthetic polymers of vinylpyrrolidone, which are used as carriers for dispersing and suspending APIs, as well as disintegrating and adhesion agents. Depending on the degree of polymerization, one can distinguish povidone—a water-soluble, linear polymer with a molecular weight from 8000 to 10,000 Daltons, and crospovidone—a water-insoluble, cross-linked vinylpyrrolidone polymer with a mass greater than 700,000 Daltons [43]. The pharmaceutical industry uses povidones with various K-values, related to the degree of polymerization, average molecular weight, and viscosity [44]. In our studies, it was found that the toxicity of povidones against *A. fischeri* increased in the series crospovidone < povidone K25 < povidone K30, with EC_50, 30 min_ values 49,855 mg/L < 1599 mg/L < 1130 mg/L, respectively (Table 1). The low toxicity of crospovidone is not a surprising result; it is a cross-linked polymer that is poorly soluble, thus exhibiting low availability for *A. fischeri*. The toxicity of povidone K25 and K30 can be considered comparable, especially for values after 5 and 15 min of exposure. The polymers are not significantly different; povidone K30 has a higher molecular weight and a higher degree of polymerization, but both forms of povidone have particles of the similar size (above 50 µm) and are characterized by complexing properties.

### 2.2. Ecotoxicity of API–Excipient and Excipient–Excipient Mixtures

The compositions of the three pharmaceutical formulations of ARBs (Valtap, Lozap, and Telmizek), studied in our previous work [10], were analyzed herein, and the ecotoxicity of two-component mixtures containing API and an excipient, present in the formulation, was examined.

Due to the large number of possible combinations, mixtures of compounds with very high EC_50_ values (as non-toxic), such as sorbitol, pregelatinized maize starch, magnesium stearate, crospovidone, and sodium stearyl fumarate, were not tested. In this study, the ecotoxicity of two-component excipient–excipient mixtures was investigated using excipients that showed interactions with APIs. The ecotoxicity of three-component mixtures (3× excipient or API + 2× excipient) was also analyzed. The ecotoxicological results for these mixtures are shown in Table 3; moreover, they are discussed in relation to the three APIs (VAL, LOS-K, TEL), and then compared with values predicted by the CA and IA models.

#### 2.2.1. Ecotoxicity of VAL–Excipient Mixtures

The experimental ecotoxicity values for mixtures containing valsartan (VAL) are presented in Table 3. Table 4 presents the EC_50, 30 min_ values which were calculated using the CA and IA models together with the calculated MDR coefficients allowing to characterize the antagonistic, additive, or synergistic effects. In this study, five binary VAL mixtures with excipients, present in the Valtap (ZENTIVA A.S.) pharmaceutical formulation, were tested.

##### Effects of Colloidal Silica

Calculations carried out with the use of both the CA and the IA models indicated the presence of synergistic effects in the mixture of VAL with colloidal silica. This was the only two-component VAL mixture in which interactions of this type were identified that made the experimental toxicity of the mixture (EC_50, 30 min_ = 56.97 mg/L) lower than the predicted toxicity (EC_50, 30 min_ = 153.5 mg/L (CA); 122.3 mg/L (IA)). It is worth recalling that the EC_50, 30 min_ values for the individual components of this mixture of VAL and colloidal anhydrous silica are 150.5 mg/L (Appendix A) and 2681 mg/L (Table 1), respectively. Knowing the physicochemical properties of both substances, it can be suspected that the presence of colloidal silica with hydrophilic properties reduces the hydrophobicity of VAL and facilitates its penetration into bacterial cells. To test, whether colloidal silica exhibits synergism with other tablet ingredients, the ecotoxicity of binary mixtures of colloidal silica with magnesium carbonate and K25 povidone were tested. The obtained results indicated antagonistic interactions in a mixture of silica with povidone K25 and additive/antagonistic interactions in the mixture with magnesium carbonate (in this case, the CA and IA models used indicated different effects). It can be suggested that the significant reduction in toxicity of the mixture of silica with povidone K25 in relation to the predicted values is due to a large molecular weight of polymeric povidone K25 and formation of silica agglomerates. The fact of agglomeration and the presence of van der Waals interactions in polymeric silica dispersions was confirmed in the scientific literature, stating that this was an obstacle that limited the use of SiO_2_ [45,46]. Interestingly, antagonistic interactions were also observed in the ternary mixture of povidone K25 + colloidal silica + magnesium carbonate, which can also be explained by the agglomeration of the components, making them less accessible to bacteria. This effect was more visible in the binary mixture (without magnesium carbonate), where the experimental EC_50, 30 min_ value was 4–11 times higher compared to the predicted value, than in the ternary mixture, where the experimental EC_50, 30 min_ value was 2–7 times higher in relation to the predicted one. Interestingly, the synergistic effect was not observed in the three-component mixture VAL + povidone K25 + colloidal silica. In this case, it might be suspected that the synergistic effect of VAL with silica cancels out the antagonistic effect between povidone K25 and silica, whereby an additive effect was observed in the ternary mixture.

##### Effects of Microcrystalline Cellulose and SLS

In binary mixtures of VAL with microcrystalline cellulose and sodium lauryl sulfate, an additive effect of the components was observed; in this case, the CA model made it possible to precisely determine the toxicity of the mixtures. In both cases, the IA model significantly underestimated the predicted toxicity values and, therefore, indicated antagonistic effects.

##### Effects of Magnesium Carbonate and Povidone K25

Both the CA and the IA models confirmed the presence of antagonistic effects in mixtures of VAL with magnesium carbonate and povidone K25. The experimental EC_50_ values of the mixtures were much higher than those calculated with the use of the above-mentioned models, which indicated that there was an effect that reduced the actual toxicity of the mixture. This effect was strongly noticeable for the VAL + magnesium carbonate mixture, where the experimental toxicity was 59 times lower than the value predicted using the CA model, which was found to be more consistent with the experimental results. This effect could be explained by the change in pH of the tested mixture; VAL is an acidic compound, while magnesium carbonate is an alkaline compound. As a result of combining these compounds, a mixture with a pH close to neutral, less harmful to *A. fischeri*, was obtained (Appendix A). Generally, it can be seen that, for VAL, the EC_50_ values predicted by the IA model were much lower than the experimental values. Hence, the calculated MDR values for the IA model more often indicated the presence of antagonistic effects.

#### 2.2.2. Ecotoxicity of LOS-K–Excipient Mixtures

The experimental ecotoxicity values for mixtures containing losartan (LOS-K) are presented in Table 3. Table 5 presents the EC_50, 30 min_ values calculated using the CA and IA models together with the calculated MDR coefficients to characterize the antagonistic, additive, or synergistic effects.

In the case of LOS-K, four binary mixtures with various excipients were tested. The smaller number of mixtures than for VAL was due to the fact that the pharmaceutical formulation of Lozap (active ingredient: LOS-K) contained fewer excipients than, for example, the formulation of Valtap (active ingredient: VAL). Interestingly, according to the assumptions of the CA model, all OS-K mixtures showed an additive effect, i.e., no increase or decrease in the mixture toxicity compared to the expected value. This was probably due to the fact that LOS-K, as the only tested ARB, was used in the form of potassium salt. This form showed better solubility and lower hydrophobicity; hence, the second component of the mixture did not significantly change the toxicity or availability of LOS-K for bacteria.

For the series of VAL and LOS-K mixtures, the values predicted using the IA model were significantly lower than the experimental values, suggesting the presence of antagonistic effects.

#### 2.2.3. Ecotoxicity of TEL–Excipient Mixtures

The experimental ecotoxicity values for mixtures containing telmisartan (TEL) are presented in Table 3. Table 6 presents the EC_50, 30 min_ values calculated using the CA and IA models together with the calculated MDR coefficients, allowing a characterization of the antagonistic, additive, or synergistic effects.

##### Effects of Mannitol and Povidone K25

In the case of TEL, the ecotoxicity of three binary TEL + excipient mixtures was tested. Both the CA and IA models allowed the identification of antagonistic effects in mixtures of TEL with mannitol and povidone K25. The observed toxicity of these mixtures was 57 and 46 times lower, respectively, than that predicted using the CA model. The differences between experimental and predicted toxicities were even greater when calculated using the IA model and amounted to 7700 and 70, respectively, while the predicted EC_50_ values using this model were again significantly lower. As with VAL, TEL also exhibited an antagonistic effect when combined with povidone K25, possibly for the same reason, i.e., agglomeration that reduced the availability of TEL to bacteria. Both TEL and VAL have hydroxyl groups in their chemical structures that can interact with the carbonyl groups of povidone K25, facilitating aggregation; such interactions are known for povidone and polyphenols [47]. The high experimental EC_50_ value for the mixture of TEL with mannitol was surprising. While the EC_50, 30 min_ values of the individual components were 77.31 mg/L (TEL, Appendix A) and 4253 mg/L (mannitol, Table 1), their mixture reached the value of 15,401 mg/L (Table 3); antagonistic effects were noticeable between TEL and mannitol, considered to be completely inert to APIs [48]. An antagonistic effect was also observed in the mixture of mannitol with povidone K25 (Table 3).

##### Effects of Meglumine

In mixtures containing meglumine (TEL + meglumine, mannitol + meglumine), according to calculations using the CA model, no antagonistic/synergistic effects were observed. On the other hand, in the mixture of two excipients, meglumine + povidone K25, strong synergistic interactions were observed (again, the IA model overestimated the predicted values). This effect could be explained by the change in pH value of the mixture in relation to the pH value of the individual substances (Appendix A). Meglumine is an alkaline compound with high toxicity toward *A. fischeri*. Povidone K25 (5% in the mixture) is a slightly toxic compound with an acidic pH of approximately 4. Meglumine predominated in the tested mixture (30% in the mixture) with a pH close to 11; hence, the pH of the mixture was alkaline, which caused cross-linking of povidone K25 and, consequently, its lower solubility. Thus, the contribution of povidone K25 to the final toxicity of the mixture was negligible [49].

#### 2.2.4. Accuracy of CA and IA Model Predictions

In conclusion, for all tested ARB mixtures (VAL, LOS-K, TEL), the use of the CA model allowed calculating the toxicity of the mixtures with a greater accuracy, whereas the IA model significantly overstated or underestimated the predicted toxicity values (Figure 2). The same finding applies to mixtures of excipients, where the predicted toxicities using the IA model were significantly overestimated. Similar results were obtained in other drug combination studies against *A. fischeri,* where the CA model was found to give a more accurate prediction of toxicity and should be recommended for modeling toxicity of a combination of drugs with different effects [50]. However, the accuracy of the prediction models used depends on the nature of the test substances; it may be the case that neither the CA nor the IA model can accurately predict the toxicity of a mixture [51]. It is worth remembering that the IA model was developed for mixtures of chemical compounds with different molecular targets, while the CA model was developed for mixtures of compounds operating according to a similar mechanism. It may be suspected that, in the studies conducted with the use of *A. fischeri*, the molecular targets of the investigated drugs were unified, and their mechanisms of toxic action were similar, usually focused on the bacterial membrane; hence, a higher precision associated with the CA model was observed.

#### 2.2.5. Ecotoxicity of Multicomponent Mixtures

In continuation of our studies, the ecotoxicity of artificially composed multicomponent mixtures was tested for all ARBs to reflect their actual pharmaceutical formulation, i.e., a mixture of all excipients, and then the same mixture with the addition of APIs (ecotoxicity of the whole tablet) (Table 3). The results are presented graphically in Figure 3. We found a trend that was already observed in our previous studies [10]; in the case of VAL, the pharmaceutical formulation (VAL + VAL excipients) was more toxic than pure VAL. The inverse dependence and a greater toxicity of APIs with respect to the pharmaceutical formulation was observed for LOS-K and TEL. This is due to the different composition of the tested formulations. The difference between the toxicity of the entire pharmaceutical formulation and its mixture of excipients was significant in previously reported studies [10]. In the case of formulations obtained by mixing individual excipients, these differences were less noticeable; they were mainly observed for LOS-K.

## 3. Materials and Methods

### 3.1. Chemicals

Pharmaceutical excipients used in this study were obtained as a gift from Polpharma SA (Starogard Gdański, Poland) (Table 7). Pure ARBs were purchased from Sigma-Aldrich (Burlington, MA, USA) as a certified reference material (Table 8). The reagents (Microtox^®^ Acute Reagent, Reconstitution Solution and Solid Phase Test Diluent) used in ecotoxicological tests were purchased from Modern Water, Inc. (New Castle, DE, USA) via the Polish distributor, Tigret LLC (Warszawa, Poland). 

### 3.2. Ecotoxicity Studies (Modified Microtox^®^ Basic Solid Phase Test)

The bioluminescent marine bacterium *Aliivibrio fischeri* (strain NRRL B-11177) was used to evaluate the acute toxicity of individual excipients, as well as the API–excipient and excipient–excipient mixtures (Figure 1). The inhibition of bioluminescence was measured using the Microtox^®^ Model 500 Analyzer (Modern Water Inc., New Castle, DE, USA). The modified Microtox^®^ Basic Solid Phase Test protocol was applied, which was described in detail in our previous work [10]; a scheme illustrating the course of the ecotoxicological study is shown in Figure 4. The bioluminescence inhibition of *A. fischeri* was measured after 5, 15, and 30 min of exposure of the bacterium to each sample and then compared with the light output of a control sample. The EC_50_ values, corresponding 95% confidence intervals, and confidence factors were estimated using the Microtox Omni 4.2. Software™.

### 3.3. Calculation of Mixtures Toxicity

The toxicity of the investigated mixtures was predicted using the concentration addition (CA) and independent action (IA) models and compared with experimental values [52]. The prediction of the combined effect of the two-component mixture based on the CA approach can be expressed as follows:(1)$${EC}_{50,mix}={\left(\frac{p_{A}}{{EC}_{50,A}}+\frac{p_{B}}{{EC}_{50,B}}\right)}^{-1}$$
where ${EC}_{50,mix}$ is the total concentration of the mixture that causes 50% inhibition of bioluminescence, $p_{A}$ is the proportion of compound A in the mixture, $p_{B}$ is the proportion of compound B in the mixture, ${EC}_{50,A}$ is the median inhibitory concentration of compound A when applied alone, and ${EC}_{50,B}$ is the median inhibitory concentration of compound B when applied alone.

The prediction of the combined effect based on the IA approach was modeled using the following formula:(2)$${E(c}_{mix})=1-\left[\left(1-e_{A}\right)\left(1-e_{B}\right)\right]$$
where ${E(c}_{mix})$ is the total effect of the mixture at a specific concentration, $e_{A}$ is the effect of compound A at that specific concentration, and $e_{B}$ is effect of compound B at that specific concentration. ${E(c}_{mix})$ was calculated from the dose–response curve determined for each, single component.

The composition of the mixtures corresponded to the composition declared by the manufacturer of a given pharmaceutical preparation; however, as pharmaceutical manufacturers reserve the right not to define the quantitative content of individual excipients, these amounts were selected in accordance with the recommendations regarding the percentage of individual excipients in the formulation.

The model deviation ratio (MDR), defined as follows, was calculated to evaluate the synergic or antagonistic effects of the tested compounds in the pharmaceutical mixtures:(3)$$MDR=\frac{{EC}_{50,mix}calculated}{{EC}_{50,mix}experimental}$$
where ${EC}_{50,mix}calculated$ is the EC_50_ value calculated using the CA/IA model (Equations (1) and (2)), and ${EC}_{50,mix}experimental$ is the EC_50_ value recorded using the Microtox^®^ test. The effects were further classified as synergistic (MDR ≥ 2), additive (0.5 < MDR < 2), or antagonistic (MDR ≤ 0.5) [53].

## 4. Conclusions

Answering the question raised in the title, it can be concluded from the research conducted in this study that pharmaceutical excipients, generally considered as safe, can threaten the aquatic environment. However, it is possible to reduce the risk. Firstly, intensive studies should be initiated into the ecotoxicity of commercial and newly marketed pharmaceutical formulations, taking into account the increase in toxicity as a result of interactions between components. Secondly, studies should be aimed at discovering new excipients, which are non-toxic to the environment. Thirdly, excipients with known, high ecotoxicity should be replaced by less toxic substitutes. Fourthly, the pharmaceutical and cosmetic industries should reduce the levels of toxic excipients in the final products. Lastly, new studies should lead to a situation where active substances can be applied in a different pharmaceutical form, omitting ecotoxic excipients, e.g., tablets containing harmful excipients could be replaced by excipient-free capsules.

Our study highlights the potential ecotoxicological hazards posed by pharmaceutical excipients, an area that has been largely overlooked. The tests carried out using bioluminescent bacteria and a recently developed modification of the Microtox^®^ test allowed estimating the ecotoxicity associated with pharmaceutical substances that can be dissolved/suspended in a liquid medium or adsorbed onto a solid environmental matrix. While some excipients may present a high risk to aquatic organisms, such as SLS and meglumine, it is essential to note that some excipients, including commonly used magnesium stearate, were found to be safe for the aquatic environment (Figure 5). Importantly, our studies confirmed the existence of a number of antagonistic and synergistic effects in the mixtures tested, including a synergistic effect between VAL and colloidal silica, as well as between meglumine and povidone K25. Moreover, the CA model allowed predicting the ecotoxicity of mixtures with a greater precision than the IA model. We observed that the ecotoxicity of certain excipients, such as silica, is influenced by factors such as particle size and commercial form, emphasizing the need for a nuanced understanding of the behavior of these materials in the environment. Our findings further reveal that the toxicity of povidones is related to their cross-linking and the degree of polymerization, which underscores the importance of evaluating specific characteristics of excipients to accurately gauge their potential environmental impact.

In conclusion, addressing the unnoticed problem of ecotoxicological hazard from pharmaceutical excipients demands concerted efforts from researchers, regulators, and pharmaceutical industries. Understanding how pharmaceutical excipients affect different organisms within the ecosystem will enable us to develop better risk assessment strategies and ensure the protection of aquatic life and the environment as a whole.

## Figures and Tables

**Figure 1 molecules-28-06590-f001:**
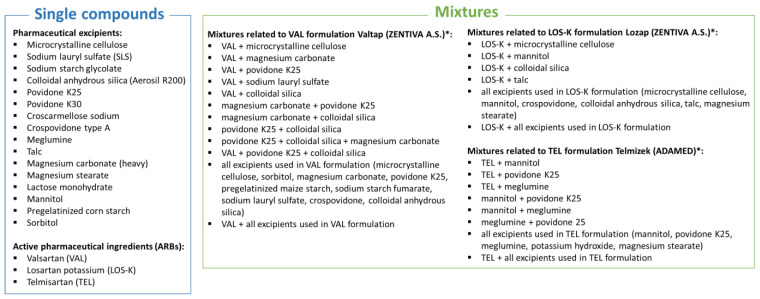
Single compounds and various types of mixtures investigated in this study. * Mixtures with APIs contain excipients that are actually present in the commercial formulations (Valtap, Lozap, Telmizek).

**Figure 2 molecules-28-06590-f002:**
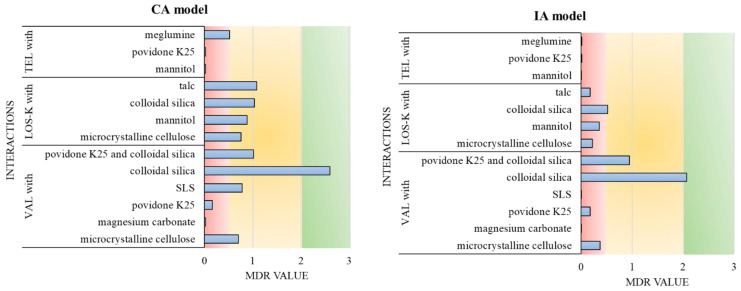
Summary of the effects observed in investigated mixtures based on the MDR values calculated using CA and IA models. Red color (MDR ≤ 0.5)—antagonistic effects, yellow color (0.5 < MDR < 2)—additive effects, and green color (MDR ≥ 2)—synergistic effects.

**Figure 3 molecules-28-06590-f003:**
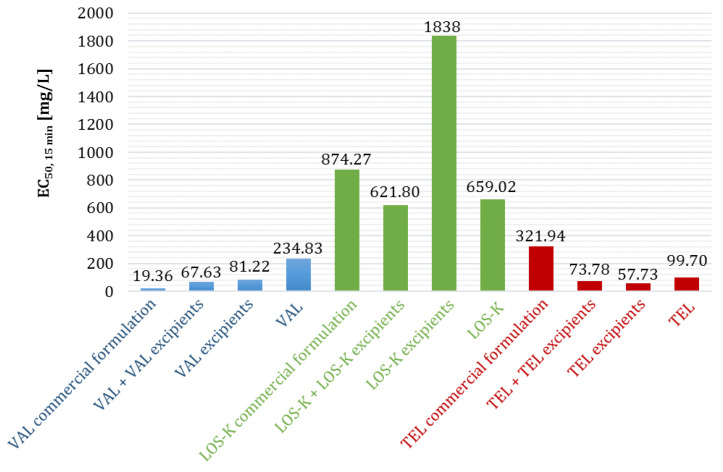
A comparison of ecotoxicity (*A. fischeri*) of the entire pharmaceutical formulation (API + API excipients) obtained by mixing individual components, ecotoxicity of the tablet mass itself (API excipients), and ecotoxicity of pure APIs with the ecotoxicity of commercial formulations [10].

**Figure 4 molecules-28-06590-f004:**
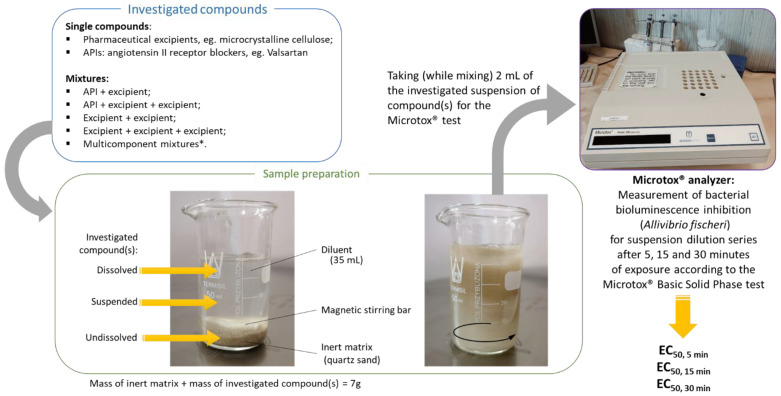
Schematic illustration of the ecotoxicological test procedure presented in this paper.

**Figure 5 molecules-28-06590-f005:**
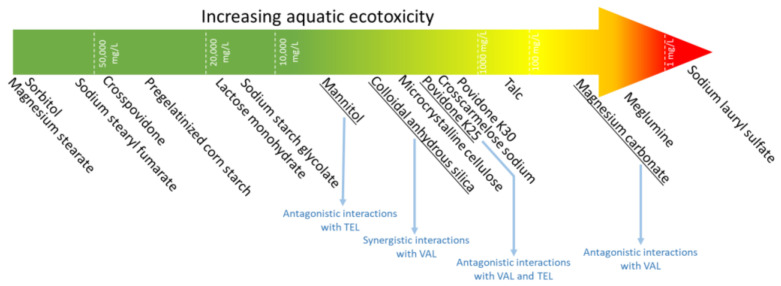
Summary of the obtained results: comparative ecotoxicity of pharmaceutical excipients along with the most important interactions with APIs.

**Table 1 molecules-28-06590-t001:** EC_50_ values of investigated pharmaceutical excipients toward *A. fischeri* obtained after 5, 15, and 30 min of exposure along with 95% confidence intervals and coefficients of determination (R^2^).

Pharmaceutical Excipient and Maximum Tested Concentration (C_max_)	EC_50_ [mg/L]; (95% Confidence Interval); R^2^			Toxicity Category [11] (Based on EC_50, 30 min_)
	5 min	15 min	30 min	
Sodium lauryl sulfate (SLS) C_max_ = 80.61 mg/L	1.371 (0.7002–2.686) 0.8893	0.8004 (0.2147–2.985) 0.7246	0.5544 (0.1138–2.702) 0.6931	I
Meglumine C_max_ = 1360 mg/L	28.56 (15.58–52.37) 0.8641	19.66 (7.958–48.57) 0.7499	18.05 (7.237–45.02) 0.7516	III
Magnesium carbonate (heavy) C_max_ = 10.15 g/L	109.8 (55.61–216.6) 0.9360	86.42 (58.99–126.6) 0.9678	65.60 (44.19–97.37) 0.9738	
Silica, nanopowder C_max_ = 3.28 g/L	100.2 (47.07–213.2) 0.8138	242.6 (141.7–415.3) 0.8882	226.4 (162.7–315.1) 0.9574	Non-toxic
Croscarmellose sodium C_max_ = 20.44 g/L	1348 (807.5–2249) 0.9163	1325 (808.0–2174) 0.9196	1400 (622.2–3150) 0.7960	
Talc C_max_ = 19.48 g/L	14,683 (305.1–706,681) 0.2736	2229 (829.2–5993) 0.6733	494.3 (196.6–1243) 0.7517	
Povidone K30 C_max_ = 15.49 g/L	2218 (1879–2619) 0.9915	1768 (405.3–7709) 0.7765	1130 (228.1–5600) 0.4681	
Povidone K25 C_max_ = 15.95 g/L	2120 (1291–3483) 0.9033	1749 (689.7–4434) 0.7225	1559 (556.7–4287) 0.7872	
Microcrystalline cellulose (type PH102) C_max_ = 2.99 g/L	1966 (742.0–5207) 0.8784	1908 (741.0–4914) 0.8827	1958 (760.8–5037) 0.8848	
Colloidal anhydrous silica (Aerosil R200) C_max_ = 7.51 g/L	3294 (1802–6021) 0.9372	3108 (1808–5344) 0.9473	2681 (1618–4443) 0.9501	
Colloidal silica, LUDOX SM-30 C_max_ = 29.50 g/L	8238 (4220–16,081) 0.9016	4776 (3078–7408) 0.9421	3794 (2481–5800) 0.9400	
Mannitol C_max_ = 14.85 g/L	12,563 (1419–111,263) 0.6267	5404 (346.9–84,181) 0.3225	4253 (209.9–86,162) 0.2195	
Sodium starch glycolate (type A) C_max_ = 12.16 g/L	10,184 (4729–21,928) 0.9539	11,725 (6045–22,743) 0.9710	12,541 ^[a]^ (5559–28,296) 0.9601	
Lactose monohydrate C_max_ = 15.08 g/L	11,705 (1477–92,754) 0.9880	13,999 (227.4–861,937) 0.9596	16,363 ^[a]^ (186.6–1,434,691) 0.9584	
Pregelatinized corn starch Cmax = 14.31 g/L	46,929 ^[a]^ (19,696–111,815) 0.9855	39,287 ^[a]^ (10,516–146,768) 0.9070	42,202 ^[a]^ (3925–453,734) 0.7540	
Crospovidone (type A) C_max_ = 14.88 g/L	27,795 ^[a]^ (274.0–2,819,364) 0.5875	64,863 ^[a]^ (9.993–421,001,866) 0.6420	49,855 ^[a]^ (35.11–70,801,223) 0.7010	
Sodium stearyl fumarate C_max_ = 5.78 g/L	56,456 ^[b]^	12,815 ^[a]^ (38.84–4,457,999) 0.9221	80,534 ^[a]^ (2479–2,616,211) 0.8896	
Sorbitol C_max_ = 31.06 g/L	Non-toxic in tested concentration, hormesis			
Magnesium stearate C_max_ = 9.57 g/L	Non-toxic in tested concentration, hormesis			

Acute toxicity categories: (I) very toxic, EC_50_ < 1.0 mg/L; (II) toxic, 1 < EC_50_ < 10; (III) harmful, 10 < EC_50_ < 100 mg/L [11]. ^[a]^ EC_50_ values calculated from extrapolated data; ^[b]^ value was calculated using two points, whereby the 95% confidence interval and R^2^ could not be determined.

**Table 2 molecules-28-06590-t002:** EC_50_ values of various forms of silica against *A. fischeri* depending on particle size.

Form of Silica	Particle Size [nm]	EC_50, 30 min_ [mg/L]	Test	Ref.
Nanopowder	10–15	226.4	A developed modification of the Microtox^®^ Basic Solid Test	This work
LUDOX SM-30 (30% colloidal solution of amorphous silica)	7	1138.2 ^[a]^		
Aerosil R200 (hydrophilic silica dust)	12	2681		
Powder	60–100	333.82	A method using a luminometer developed by the authors	[30]
Mesoporous silica	60–150	319.68	The Microtox^®^ Basic Test for extracts	[31]

^[a]^ Calculated in reference to pure SiO_2._

**Table 3 molecules-28-06590-t003:** EC_50_ values of investigated mixtures containing excipients toward *A. fischeri* obtained after 5, 15, and 30 min of exposure along with 95% confidence intervals and coefficient of determination (R^2^).

API	Mixture and Maximum Tested Concentration (C_max_)	Composition (wt.%) of the Mixture	EC_50_ [mg/L]; (95% Confidence Interval); R^2^		
			5 min	15 min	30 min
VAL	Mixtures: VAL + excipient				
	VAL + microcrystalline cellulose C_max_ = 8.75 g/L	47% + 23%	319.0 (132.3–769.0) 0.7959	382.5 (103.6–1412) 0.6419	307.9 (84.13–1127) 0.5859
	VAL + magnesium carbonate C_max_ = 3.22 g/L	47% + 5%	24,619 ^[a]^ (2674–226,629) 0.9616	15,105 ^[a]^ (1060–215,266) 0.7598	7855 ^[a]^ (1295–47,651) 0.7972
	VAL + povidone K25 C_max_ = 4.29 g/L	47% + 5%	675.2 (371.7–1226) 0.8966	571.3 (289.3–1128) 0.8580	1046 (158.5–6907) 0.9717
	VAL + sodium lauryl sulfate C_max_ = 3.96 g/L	47% + 1%	34.75 (29.38–41.10) 0.9936	29.39 (22.48–38.41) 0.9898	28.91 (19.82–42.16) 0.9802
	VAL + colloidal silica C_max_ = 3.96 g/L	47% + 1%	130.7 ^[a]^ (128.3–133.2) 1.0000	111.6 (49.01–254.2) 0.9937	58.97 (39.42–88.23) 0.9747
	Mixtures: excipient + excipient				
	Magnesium carbonate + povidone K25 Cmax = 24.75 g/L	5% + 5%	240.6 (154.5–374.6) 0.9557	154.8 (59.19–404.6) 0.8587	85.99 ^[a]^ (9.907–746.4) 0.7819
	Magnesium carbonate + colloidal silica Cmax = 9.90 g/L	5% + 1%	114.1 ^[a]^ (24.14–539.1) 0.7982	118.3 (83.90–166.8) 0.9662	71.10 ^[a]^ (4.159–1215) 0.6047
	Povidone K25 + colloidal silica Cmax = 19.80 g/L	5% + 1%	11 028 (1181–102,957) 0.7085	5845 (2560–13,343) 0.7972	6199 (3114–12,342) 0.8622
	Ternary mixtures				
	Povidone K25 + colloidal silica + magnesium carbonate Cmax = 18.15 g/L	5% + 1% + 5%	553.4 (385.5–794.4) 0.9563	353.1 (206.4–600.7) 0.9238	305.8 (171.3–546.0) 0.8825
	VAL + povidone K25 + colloidal silica Cmax = 4.37 g/L	47% + 5% + 1%	258.4 (115.0–580.4) 0.7962	179.6 (66.95–482.0) 0.7297	163.8 (74.04–362.3) 0.8299
	Multicomponent mixtures				
	VAL excipients ^[b]^ Cmax = 1.58 g/L	23% + 12% + 5% + 5% + 5% + 2% + 1% + 2% + 1%	164.4 (131.0–206.3) 0.9797	81.22 (69.57–94.82) 0.9915	59.26 (53.52–65.50) 0.9964
	VAL + VAL excipients ^[b]^ Cmax = 2.91 g/L	47% + 23% + 12% + 5% + 5% + 5% + 2% + 1% + 2% + 1%	129.2 (107.9–154.6) 0.9880	67.63 (56.89–80.40) 0.9915	47.85 (35.84–63.89) 0.9819
LOS-K	Mixtures: LOS-K + excipient				
	LOS-K + microcrystalline cellulose Cmax = 10.23 g/L	24% + 38%	931.2 (809.2–1072) 0.9945	770.0 (167.0–3550) 0.9752	771.6 (79.31–7506) 0.9457
	LOS-K + mannitol Cmax = 7.26 g/L	24% + 20%	691.8 (637.7–750.6) 0.9981	618.2 (530.2–720.8) 0.9937	546.6 (440.5–678.4) 0.9887
	LOS-K + colloidal silica Cmax = 4.12 g/L	24% + 1%	391.8 (279.7–548.8) 0.9876	294.1 (275.9–313.5) 0.9992	278.8 (227.9–341.2) 0.9971
	LOS-K + talc Cmax = 4.12 g/L	24% + 1%	440.7 (327.2–593.6) 0.9733	319.7 (232.4–440.0) 0.9717	260.3 (180.9–374.5) 0.9479
	Multicomponent mixtures				
	LOS-K excipients ^[c]^ Cmax = 10.89 g/L	38% + 20% + 5% + 1% + 1% + 1%	1519 (1300–1774) 0.9914	1838 (1454–2324) 0.9796	2101 (1631–2708) 0.9763
	LOS-K + LOS-K excipients ^[c]^ Cmax = 14.85 g/L	24% + 38% + 20% + 5% + 1% + 1% + 1%	800.4 (475.0–1349) 0.9140	621.8 (336.9–1148) 0.8510	536.9 (268.5–1074) 0.8129
TEL	Mixtures: TEL + excipient				
	TEL + mannitol Cmax = 10.23 g/L	17% + 45%	13,602 ^[a]^ (46.14–4,009,769) 0.8602	13,882 ^[a]^ (3052–63,153) 0.9893	15,401 ^[a]^ (2575–92,096) 0.9872
	TEL + povidone K25 Cmax = 13.36 g/L	17% + 5%	4017 (3018–5347) 0.9859	4583 (3454–6080) 0.9856	4566 (3455–6034) 0.9861
	TEL + meglumine Cmax = 3.87 g/L	17% + 30%	66.42 (27.66–159.5) 0.7661	53.11 (9.586–294.2) 0.5387	48.05 (6.499–355.2) 0.3256
	Mixtures: excipient + excipient				
	Mannitol + povidone K25 Cmax = 8.25 g/L	45% + 5%	9620 ^[a]^ (355.2–260,504) 0.6879	34 023 ^[a]^ (42.72–27,097,494) 0.9664	no toxicity
	Mannitol + meglumine Cmax = 6.19 g/L	45% + 30%	136.2 (26.70–694.9) 0.9633	83.17 (26.37–262.2) 0.7808	73.14 (20.99–254.9) 0.7226
	Meglumine + povidone 25 Cmax = 0.26 g/L	30% + 5%	1.058 ^[a]^ (0.5888–1.901) 0.9588	0.9118 (0.6509–1.277) 0.9879	1.011 ^[a]^ (0.7103–1.440) 0.9856
	Multicomponent mixtures				
	TEL excipients ^[d]^ Cmax = 6.85 g/L	45% + 5% + 30% + 2% + 1%	85.27 (46.67–155.8) 0.8943	57.73 (17.69–188.4) 0.7822	48.25 (13.97–166.6) 0.8052
	TEL + TEL excipients ^[d]^ Cmax = 4.12 g/L	17% + 45% + 5% + 30% + 2% + 1%	75.47 (11.90–478.8) 0.7019	73.78 (3.562–1528) 0.2907	50.19 (6.503–387.3) 0.2873

^[a]^ EC_50_ values calculated from extrapolated data; ^[b]^ VAL excipients (Valtap): microcrystalline cellulose, sorbitol, magnesium carbonate, povidone K25, pregelatinized maize starch, sodium starch fumarate, sodium lauryl sulfate, crospovidone, colloidal anhydrous silica; ^[c]^ LOS-K excipients (Lozap): microcrystalline cellulose, mannitol, crospovidone, colloidal anhydrous silica, talc, magnesium stearate; ^[d]^ TEL excipients (Telmizek): mannitol, povidone K25, meglumine, potassium hydroxide, magnesium stearate.

**Table 4 molecules-28-06590-t004:** Predicted toxicities of VAL–excipient and excipient–excipient mixtures calculated using the CA and IA models together with MDR values.

Mixture	EC_50, 30 min_ [mg/L]			MDR		Predicted Effect	
	Experimental	CA	IA	CA	IA	CA	IA
VAL + microcrystalline cellulose	307.9	216.0	114.1	0.70	0.37	Additive	Antagonistic
VAL + magnesium carbonate	7855	133.8	16.54	0.02	0.002	Antagonistic	Antagonistic
VAL + povidone K25	1046	164.8	184.1	0.16	0.18	Antagonistic	Antagonistic
VAL + sodium lauryl sulfate	28.91	22.68	0.06	0.78	0.002	Additive	Antagonistic
VAL + colloidal silica	58.97	153.5	122.3	2.60	2.07	Synergistic	Synergistic
Magnesium carbonate + povidone K25	85.99	125.9	55.59	1.46	0.65	Additive	Additive
Magnesium carbonate + colloidal silica	71.10	78.33	18.11	1.10	0.25	Additive	Antagonistic
Povidone K25 + colloidal silica	6199	1676	530.2	0.27	0.08	Antagonistic	Antagonistic
Povidone K25 + colloidal silica + magnesium carbonate	305.8	137.8	44.0	0.45	0.14	Antagonistic	Antagonistic
VAL + povidone K25 + colloidal silica	163.8	167.8	155.17	1.02	0.95	Additive	Additive

Classification of effects: synergistic (MDR ≥ 2), additive (0.5 < MDR < 2), antagonistic (MDR ≤ 0.5).

**Table 5 molecules-28-06590-t005:** Predicted toxicities of LOS-K–excipient and excipient–excipient mixtures calculated using the CA and IA models together with MDR values.

Mixture	EC_50, 30 min_ [mg/L]			MDR		Predicted Effect	
	Experimental	CA	IA	CA	IA	CA	IA
LOS-K + microcrystalline cellulose	771.6	584.1	167.3	0.76	0.22	Additive	Antagonistic
LOS-K + mannitol	546.6	481.2	197.5	0.88	0.36	Additive	Antagonistic
LOS-K + colloidal silica	278.8	287.0	143.7	1.03	0.52	Additive	Additive
LOS-K + talc	260.3	281.7	46.61	1.08	0.18	Additive	Antagonistic

Classification of effects: synergistic (MDR ≥ 2), additive (0.5 < MDR < 2), antagonistic (MDR ≤ 0.5).

**Table 6 molecules-28-06590-t006:** Predicted toxicities of TEL–excipient and excipient–excipient mixtures calculated using the CA and IA models together with MDR values.

Mixture	EC_50, 30 min_ [mg/L]			MDR		Predicted Effect	
	Experimental	CA	IA	CA	IA	CA	IA
TEL + mannitol	15,401	269.01	2.02	0.02	0.0001	Antagonistic	Antagonistic
TEL + povidone K25	4566	98.61	64.81	0.02	0.01	Antagonistic	Antagonistic
TEL + meglumine	48.05	24.97	0.54	0.52	0.01	Additive	Antagonistic
Mannitol + povidone K25	34,023 ^[a]^	3626.35	815.81	0.11	0.02	Antagonistic	Antagonistic
Mannitol + meglumine	73.14	44.84	17.97	0.61	0.25	Additive	Antagonistic
Meglumine + povidone K25	1.011	21.02	41.71	20.79	41.26	Synergistic	Synergistic

Classification of effects: synergistic (MDR ≥ 2), additive (0.5 < MDR < 2), antagonistic (MDR ≤ 0.5). ^[a]^ Sample was non-toxic with an exposure time of 30 min; instead, the EC_50, 15 min_ value is shown.

**Table 7 molecules-28-06590-t007:** Chemical structures of pharmaceutical excipients investigated in this paper.

Pharmaceutical Excipient	Chemical Structure
Microcrystalline cellulose type PH102	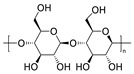
Sodium lauryl sulfate (SLS)	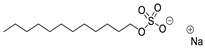
Sodium starch glycolate	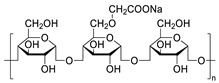
Colloidal anhydrous silica (Aerosil R200)	[SiO_2_]_n_
Povidone K25	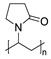 Molecular weight ~24,000 g
Povidone K30	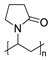 Molecular weight ~40,000 g
Croscarmellose sodium	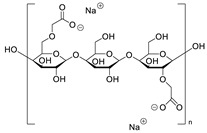
Crospovidone type A	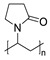 Cross-linked polymer
Meglumine	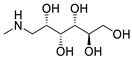
Talc	Mg_3_(OH)_2_Si_4_O_10_
Magnesium carbonate (heavy)	MgCO_3_ + Mg(OH)_2_
Magnesium stearate	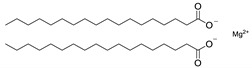
Lactose monohydrate	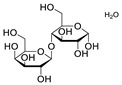
Mannitol	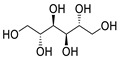
Pregelatinized corn starch	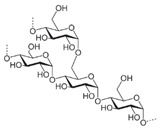
Sorbitol	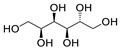

**Table 8 molecules-28-06590-t008:** Chemical structures of ARBs used in this study.

ARB	Chemical Structure
Valsartan (VAL)	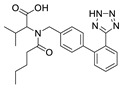
Losartan potassium (LOS-K)	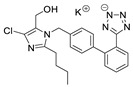
Telmisartan (TEL)	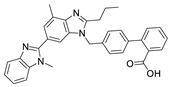

## Data Availability

Data are contained within the article or Appendix A.

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
