# Peer review of "Can Pharmaceutical Excipients Threaten the Aquatic Environment? A Risk Assessment Based on the Microtox® Biotest"

_molecules, 2023, doi:10.3390/molecules28186590_

Round 1
Reviewer 1 Report
This study sheds light on a very important issue, the ecotoxicological impact of excipients. The potential risks that pharmaceutical excipients may pose to the aquatic environment was evaluated using the Microtox test.
However, the manuscript is too long and very difficult to read and understand (the methodology and the results) , the information is not given in a straight forward manner. I advise the authors to shorten some parts where possible and to give at least two schematic illustration fugure one for procedures to summarize the methodology. The concusion should be also corrected to be more specific.
The titles implies that the manuscript is a review article and not a research article, the type of the study should be clearly mentioned. Also the titles is very general, the author should clearly mention the category of the tested excipients, for example “tablet excipients, or the most used tablet excipients or diluents ” , the title is the first sentence the guide the reader and help him to expect what he is going to read.
Below are my comments and recommendations:
The title gives an impression that the paper is a review article and not a research article, therefore I recommend to add the type of study to the title.
Define the Microtox test in the first time its mentioned in the introduction
Line 67: should be a new paragraph
Lines 91-92: Explain the basic difference between the two methos , CA and IA and why id is important to carry out them both
Line 122, a typo : The toxicity of the investigated mixtures were predicted, should be was and not were
Tables 3 and 5 are extremely confusing, consider adding borders between the rows or at least space between the values for each tested excipient or excipients mixtures
I strongly suggest building a table at the end of section 3.1. to clearly summarize (in words) the results and the toxicity of each tested excipients. It should not contain numbers, rather a translation of the number in Table 3.
Surfactants other than SLS are not studied in this work. Although it is very interesting to know about their ecotoxicological effects relative to SLS.
Moderate English editing is required
Author Response
Thank you very much for taking the time to review this manuscript. Please find the detailed response in attachment file and the corresponding revisions highlighted in the re-submitted files.

Reviewer 2 Report
This study aims to evaluate the potential risks that pharmaceutical excipients, generally considered safe, may pose to the aquatic environment. The authors used the concentration addition (CA) and independent action (IA) models to shed light on the often overlooked ecotoxicological consequences of these substances. Furthermore, they used modification of the Microtox® Basic Solid Phase protocol developed for poorly-soluble substances. The results obtained suggest high toxicity of some excipients, i.e. SLS and meglumine, and confirm the occurrence of interactions between API and excipients.
The topics discussed in the article are interesting, but there are some flaws in the description of the parts and the format of the article. Therefore, I can only recommend that this article be published on the molecules after the following revisions:
1. There are many red traces of modification in the article, please check carefully. For example, in the summary section and in the pages 10, Line 319 and 326.
2. The three-line table in Table 2 in the article is missing.
3. Some of the subheadings in the third section of the article are missing serial numbers.
4. The format of the references in the article is as uniform as possible.
5. Page 7, Line 186~188. “With regard to ...... so this compound can be classified to the toxicity class II. Here the authors describe SLS as classified as Class II toxicity, but why is it written in the table about Class I toxicity?
6. Articles pages 7, Line 206~208. “In the course ...... its EC50 value is over 31061
mg/L...... "Where is the 31061 mg/L data ?
7. Page 13, Line 351~352. “VAL and colloidal silica are respectively: 150.5 mg/L (Supplementary Material, Table S2) and 2681 mg/L (Table 3). ” However, the EC50, 30 min for colloidal silica listed in Table 3 is 3794 mg/L. Please check carefully.
8. Page 15, Line 438~440. “While the EC50, 15 min values of...... their mixture reached the value of 15401 mg/L (Table 5) ..." The EC50, 15 min =15401 mg/L is inconsistent with the EC50, 15 min =13882 mg/L given in Table 5.
Minor editing of English language required
Author Response
Thank you very much for taking the time to review this manuscript. Please find the detailed responses in attachment file and the corresponding revisions highlighted in the re-submitted files.

Round 2
Reviewer 1 Report
Significantly improved
MInor english editing are needed